# Bayesian inference of COVID-19 spreading rates in South Africa

Rendani Mbuvha[1,2]*, Tshilidzi Marwala[2]

**1** School of Statistics and Actuarial Science, University of Witwatersrand, Johannesburg, South Africa,
**2** Institute of Intelligent Systems, University of Johannesburg, Johannesburg, South Africa

* rendani.mbuvha@wits.ac.za

## Abstract

The Severe acute respiratory syndrome coronavirus 2 (SARS-CoV-2) pandemic has highlighted the need for performing accurate inference with limited data. Fundamental to the design of rapid state responses is the ability to perform epidemiological model parameter inference for localised trajectory predictions. In this work, we perform Bayesian parameter inference using Markov Chain Monte Carlo (MCMC) methods on the Susceptible-Infected-Recovered (SIR) and Susceptible-Exposed-Infected-Recovered (SEIR) epidemiological models with time-varying spreading rates for South Africa. The results find two change points in the spreading rate of COVID-19 in South Africa as inferred from the confirmed cases. The first change point coincides with state enactment of a travel ban and the resultant containment of imported infections. The second change point coincides with the start of a state-led mass screening and testing programme which has highlighted community-level disease spread that was not well represented in the initial largely traveller based and private laboratory dominated testing data. The results further suggest that due to the likely effect of the national lockdown, community level transmissions are slower than the original imported case driven spread of the disease.

**Data Availability Statement:** The data underlying the results presented in the study are available from the John Hopkins Coronavirus resource center. URL: https://coronavirus.jhu.edu/map.html.

**Funding:** RM is supported by the Google PhD fellowship Programme. URL:

## Introduction

The first reported case of the novel coronavirus (SARS-CoV-2) in South Africa was announced on 5 March 2020, following the initial manifestation of the virus in Wuhan China in December 2019 [1–3]. Due to its further spread and the severity of its associated clinical outcomes, the disease was subsequently declared a pandemic by the World Health Organisation (WHO) on 11 March 2020 [1, 2]. In South Africa, by 26 April 2020, 4546 people had been confirmed to have been infected by the coronavirus with 87 fatalities [4].

Numerous states have attempted to minimise the growth in number of COVID-19 infections [1, 5, 6]. These attempts are largely based on non-pharmaceutical interventions (NPIs) aimed at separating the infectious population from the susceptible population [1].

These initiatives aim to strategically reduce the increase in infections to a level where their healthcare systems stand a chance of minimising the number of fatalities [1]. Some of the critical indicators for policymaker response planning include projections of the infected population,

https://research.google/outreach/phd-fellowship/.
The funders had no role in study design, data
collection and analysis, decision to publish, or
preparation of the manuscript. No additional
funding was received for this study.

**Competing interests:** The authors have declared
that no competing interests exist.

estimates of health care service demand and whether current containment measures are effective [1].

As the pandemic develops in a rapid and varied manner in most countries, calibration of epidemiological models based on available data can prove to be [7]. This difficulty is further escalated by the high number of asymptomatic cases and the limited testing capacity [1, 2].

A fundamental issue when calibrating localised models is inferring parameters of compartmental models such as susceptible-infectious-recovered (SIR) and the susceptible-exposed-infectious-recovered (SEIR) that are widely used in infectious disease projections. In the view of public health policymakers, a critical aspect of projecting infections is the inference of parameters that align with the underlying trajectories in their jurisdictions. The spreading rate is a parameter of particular interest which is subject to changes due to voluntary social distancing measures and government-imposed contact bans.

The uncertainty in utilising these models is compounded by the limited data in the initial phases and the rapidly changing dynamics due to rapid public policy changes.

To address these complexities, we utilise the Bayesian Framework for the inference of epidemiological model parameters in South Africa. The Bayesian framework allows for both incorporation of prior knowledge and principled embedding of uncertainty in parameter estimation.

In this work we combine Bayesian inference with the compartmental SEIR and SIR models to infer time varying spreading rates that allow for quantification of the impact of government interventions in South Africa.

## Methods

### Epidemiological modelling

Compartmental models are a class of models that is widely used in epidemiology to model transitions between various stages of disease [1, 8, 9]. We now introduce the Susceptible-Exposed-Infectious-Recovered (SEIR) and the related Susceptible-Infectious-Recovered (SIR) compartmental models that have been dominant in COVID-19 modelling literature [1, 5, 6, 10].

**The Susceptible-Exposed-Infectious-Recovered Model.** The SEIR is an established epidemiological model for the projection of infectious diseases. The SEIR models the transition of individuals between four stages of a condition, namely:

- being susceptible to the condition,

- being infected and in incubation

- having the condition and being infectious to others and

- having recovered and built immunity for the disease.

The SEIR can be interpreted as a four-state Markov chain which is illustrated diagrammatically in Fig 1. The SEIR relies on solving the system of ordinary differential equations below

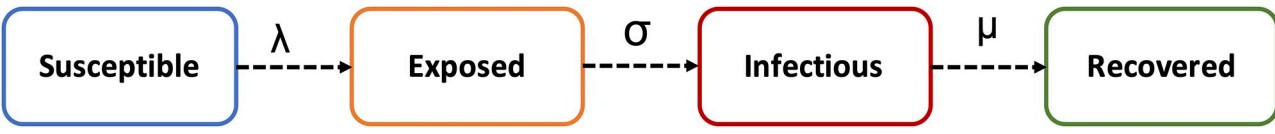

**Fig 1. An Illustration of the underlying states of the Susceptible-Exposed-Infectious-Recovered Model(SEIR).**

representing the analytic trajectory of the infectious disease [1].

$$\frac{dS}{dt} = -\frac{\lambda SI}{N} \tag{1}$$

$$\frac{dE}{dt} = \frac{\lambda SI}{N} - \sigma E \tag{2}$$

$$\frac{dI}{dt} = \sigma E - \mu I \tag{3}$$

$$\frac{dR}{dt} = \mu I \tag{4}$$

Where S is the susceptible population, I is the infected population, R is the recovered population and N is the total population where $N = S + E + I + R$. $\lambda$ is the transmission rate, $\sigma$ is the rate at which individuals in incubation become infectious, and $\mu$ is the recovery rate. $1/\sigma$ and $1/\mu$ therefore, become the incubation period and contagious period respectively.

We also consider the Susceptible-Infectious-Recovered (SIR) model which is a subclass of the SEIR model that assumes direct transition from the susceptible compartment to the infected (and infectious) compartment. The SIR is represented by three coupled ordinary differential equations rather than the four in the SEIR. Fig 2 depicts the three states of the SIR model.

**The basic reproductive number $R_0$.** The basic reproductive number ($R_0$) represents the mean number of additional infections created by one infectious individual in a susceptible population. According to the latest available literature, without accounting for any social distancing policies the $R_0$ for COVID-19 is between 2 and 3.5 [2, 6, 10, 11]. $R_0$ can be expressed in terms of $\lambda$ and $\mu$ as:

$$R_0 = \frac{\lambda}{\mu} \tag{5}$$

**Extensions to the SEIR and SIR models.** We use an extended version of the SEIR and SIR models of [6] that incorporates some of the observed phenomena relating to COVID-19. First we include a delay $D$ in becoming infected ($I^{\text{new}}$) and being reported in the confirmed case statistics, such that the confirmed reported cases $CR_t$ at some time $t$ are in the form [6]:

$$CR_t = I^{\text{new}}_{t-D} \tag{6}$$

We further assume that the spreading rate $\lambda$ is time-varying rather than constant with change points that are affected by government interventions and voluntary social distancing measures.

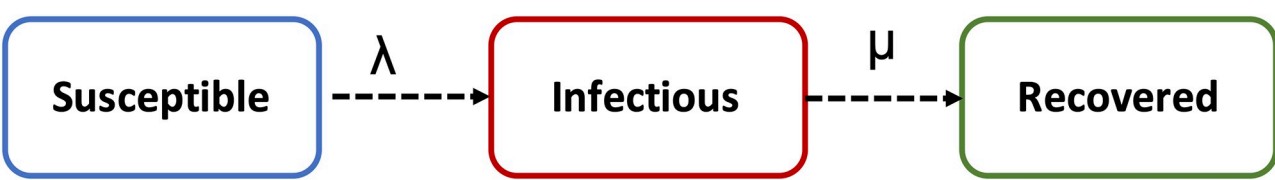

**Fig 2. An Illustration of the underlying states of the Susceptible-Infectious-Recovered Model(SIR).**

## Bayesian parameter inference

We follow the framework of [6] to perform Bayesian inference for model parameters on the South African COVID-19 data. The Bayesian framework allows for the posterior inference of parameters which updates prior beliefs based on a data-driven likelihood. The posterior inference is governed by Bayes theorem as follows:

$$P(W|D, M) = \frac{P(D|W, M)P(W)}{P(D)} \qquad (7)$$

Where $P(W|D, M)$ is the posterior distribution of a vector of model parameters ($W$) given the model($M$) and observed data($D$), $P(D|W, M)$ is the data likelihood and $P(D)$ is the evidence.

**The likelihood.** The Likelihood indicates the probability of observing the reported case data given the assumed model. In our study, we adopt the Student-T distribution as the Likelihood as suggested by [6]. Similar to a Gaussian likelihood, the Student-T likelihood allows for parameter updates that minimise discrepancies between the predicted and observed reported cases.

**Priors.** Parameter prior distributions encode some prior subject matter knowledge into parameter estimation. In the case of epidemiological model parameters, priors incorporate literature based expected values of parameters such as recovery rate($\mu$), spreading rate($\lambda$), change points based on policy interventions etc.

The prior settings for the model parameters are listed in Table 1. We follow [6] by selecting LogNormal distributions for $\lambda$ and $\sigma$ such that the initial mean basic reproductive number is 3.2 which is consistent with literature [2, 5, 6, 10, 12]. We set a LogNormal prior for the $\sigma$ such that the mean incubation period is five days. We use the history of government interventions to set priors on change points in the spreading rate. The priors on change-points include 19/03/2020 when a travel ban and school closures were announced, and 28/03/2020 when a national lockdown was enforced. We keep the priors for the Lognormal distributions of the spreading rates after the change points weakly-informative by setting the same mean as $\lambda_0$ and higher variances across all change points. This has the effect of placing greater weight on the data driven likelihood. Similar to [6] we adopt weakly-informative Half-Cauchy priors for the initial conditions for the infected and exposed populations.

**Markov Chain Monte Carlo (MCMC).** Given that the closed-form inference of the posterior distributions on the parameters listed in Table 1 is infeasible, we make use of Markov Chain Monte Carlo to sample from the posterior. Monte Carlo methods approximate solutions

**Table 1. Prior distribution settings for SEIR and SIR model parameters.**

| Parameter | Prior Distribution |
| --- | --- |
| Spreading rate $\lambda_0$ | LogNormal(log(0.4),0.5) |
| Spreading rate $\lambda_1$ | LogNormal(log(0.4),0.7) |
| Spreading rate $\lambda_2$ | LogNormal(log(0.4),0.7) |
| Incubation to infectious rate $\sigma$ | LogNormal(log(1/5),0.5) |
| Recovery rate $\mu$ | LogNormal(log(1/8),0.2) |
| Reporting Delay $D$ | LogNormal(log(8),0.2) |
| Initial Infectious $I_0$ | Half-Cauchy(20) |
| Initial Exposed $E_0$ | Half-Cauchy(20) |
| Change Point $t_1$ | Normal(2020/03/18,1) |
| Change Point $t_2$ | Normal(2020/03/28,1) |

to complex numerical problems by simulating a random process. MCMC uses a Markov Chain to sample from the posterior distribution, where a Markov Chain is a sequence of random variables $W_t$ such that:

$$P(W_{t+1}|W_1, ..., W_t) = P(W_{t+1}|W_t)$$

MCMC techniques have been widely used in COVID-19 parameter inference [6, 10]. In this work, we explore inference using Metropolis-Hastings (MH), Slice Sampling and No-U-Turn Sampler (NUTS).

**Metropolis Hastings (MH).** MH is one of the simplest algorithms for generating a Markov Chain which converges to the correct stationary distribution. The MH generates proposed samples using a proposal distribution. A new parameter state $W_{t^*}$ is accepted or rejected probabilistically based on the posterior likelihood ratio:

$$P(accept(W_{t*})) = min\left(1, \frac{P(W_{t*}|D, M)}{P(W^{t-1}|D, M)}\right) \qquad (8)$$

A common proposal distribution is a symmetric random walk obtained by adding Gaussian noise to a previously accepted parameter state. Random walk behaviour of such a proposal typically results in low sample acceptance rates.

**Slice sampling.** Slice sampling facilitates sampling from the posterior distribution $P(W|D, M)$ by adding an auxiliary variable $u$ such that the joint posterior distribution becomes:

$$P(W, u|D, M) = \begin{cases} \frac{1}{Z} & 0 \leq U \leq P(W|D, M) \\ 0 & \text{Otherwise} \end{cases} \qquad (9)$$

Where $Z = \int P(W|D, M)dW$ which is a normalisation constant. Marginal samples for the parameters W can then be obtained by ignoring $u$ samples from the joint samples. This process corresponds to sampling above the slice of the posterior density function around a predefined window. Fig 3 shows an illustration of slice sampling.

While sample acceptance is guaranteed with slice sampling, a large slice window can lead to computationally inefficient sampling while a small window can lead to poor mixing.

**Hybrid Monte Carlo (HMC) and the No-U-Turn Sampler (NUTS).** Metropolis-Hastings (MH) and slice sampling tend to exhibit excessive random walk behaviour—where the next state of the Markov Chain is randomly proposed from a proposal distribution [13–15]. This results in low proposal acceptance rates and small effective sample sizes.

HMC proposed by [16] reduces random walk behaviour by adding auxiliary momentum variables to the parameter space [15]. HMC creates a vector field around the current state using gradient information, which assigns the current state a trajectory towards a high probability next state [15]. The dynamical system formed by the model parameters $W$ and the auxiliary momentum variables $p$ is represented by the Hamiltonian $H(W, p)$ written as follows [15, 16]:

$$H(W, p) = M(W) + K(p) \qquad (10)$$

Where $M(W)$ is the negative log-likelihood of the posterior distribution in Eq 7, also referred to as the potential energy. $K(p)$ is the kinetic energy defined by the kernel of a

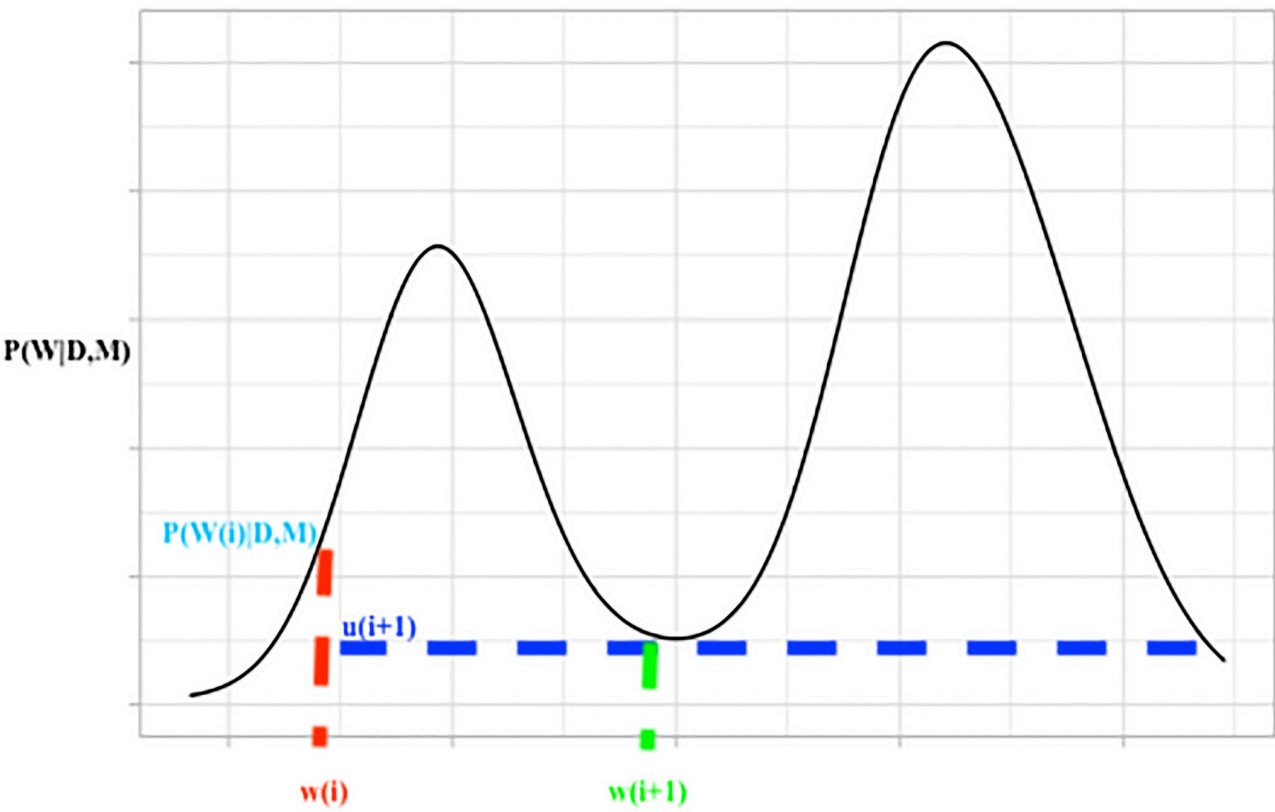

**Fig 3. An illustration of slice sampling, moving from a parameter sample w(i) to w(i + 1) via auxiliary variable sample u(i + 1).**

Gaussian with a covariance matrix $M$ [17]:

$$K(p) = \frac{p^T M^{-1} p}{2}.$$  (11)

The trajectory vector field is defined by considering the parameter space as a physical system that follows Hamiltonian Dynamics [15]. The dynamical equations governing the trajectory of the chain are then defined by Hamiltonian equations at a fictitious time $t$ as follows [16]:

$$\frac{\partial w_i}{\partial t} = \frac{\partial H}{\partial p_i}$$  (12)

$$\frac{\partial p_i}{\partial t} = -\frac{\partial H}{\partial w_i}$$  (13)

In practical terms, the dynamical trajectory is discretised using the leapfrog integrator. In the leapfrog integrator to reach the next point in the path, we take half a step in the momentum direction, followed by a full step in the direction of the model parameters—then ending with another half step in the momentum direction.

Due to the discretising errors arising from leapfrog integration a Metropolis acceptance step is then performed in order to accept or reject the new sample proposed by the trajectory [15, 18]. In the Metropolis step the parameters proposed by the HMC trajectory $w^*$ are

accepted with the probability [16]:

$$P(accept) = \min\left(1, \frac{P(w^*|D, \alpha, \beta, H)}{P(w|D, \alpha, \beta, H)}\right) \tag{14}$$

Algorithm 1 shows the pseudo-code for the HMC where $\epsilon$ is a discretisation stepsize. The leapfrog steps are repeated until the maximum trajectory length $L$ is reached.

**Algorithm 1**: Hybrid Monte Carlo Algorithm

**Data:** Confirmed Cases dataset $\{\mathbf{C}^{(t)}\}$
**Result:** *N* Samples of model parameters **W**
**for** *n* ← 1 **to** *N* **do**
 $w_0$ ← $w_{\text{init}}$
 *sample the auxiliary momentum variables p*
 p ∼ $\mathcal{N}(0, \mathbf{M})$
 Use leapfrog steps to generate proposals for *w*
 **for** *t* ← 1 **to** *L* **do**

$$p(t + \epsilon/2) \leftarrow p(t) + (\epsilon/2)\frac{\partial H}{\partial w}(w(t))$$

$$w(t + \epsilon) \leftarrow w(t) + \epsilon\frac{p(t + \epsilon/2)}{M}$$

$$p(t + \epsilon) \leftarrow p(t + \epsilon/2) + (\epsilon/2)\frac{\partial H}{\partial w}(w(t + \epsilon))$$

 **end**
 *Metropolis Update step*:
 (p, w)$_n$ ← (p(L), w(L)) with probability:

$$min\left(1, \frac{P(w_{t*}|D, M)}{P(w|D, M)}\right)$$

**end**

The HMC algorithm has multiple parameters that require tuning for efficient sampling, such as the step size and the trajectory length. In terms of trajectory length, a trajectory length that is too short leads to random walk behaviour similar to MH. While a trajectory length that is too long results in a trajectory that inefficiently traces back.

The stepsize is also a critical parameter for sampling, small stepsizes are computationally inefficient leading to correlated samples and poor mixing while large stepsizes compound discretisation errors leading to low acceptance rates. Tuning these parameters requires multiple time consuming trial runs.

NUTS automates the tuning of the leapfrog stepsize and trajectory length. In NUTS the stepsize is tuned during an initial burn-in phase by targeting particular levels of sample acceptance. The trajectory length is tuned by iteratively adding steps until either the chain starts to trace back (U-turn) or the Hamiltonian explodes (becomes infinite).

**Table 2. Leave-one out (LOO) statistics comparing SEIR and SIR models with different number of change points.**

| Model | Change Points | LOO | Effective Parameters |
|-------|---------------|-----|----------------------|
| SIR | 2 | 448.00 | 10.27 |
| SEIR | 1 | 457.77 | 11.60 |
| SEIR | 2 | 459.94 | 12.00 |
| SIR | 1 | 463.03 | 8.51 |
| SEIR | 0 | 464.69 | 16.14 |
| SIR | 0 | 517.72 | 4.72 |

We use the samplers described above to calibrate the SEIR and SIR models on daily new cases and cumulative cases data for South Africa up to and including 20 April 2020 provided by Johns Hopkins University's Center for Systems Science and Engineering(CSSE) [3].

## Results

SIR and SEIR model parameter inference was performed using confirmed cases data up to and including 20 April 2020 and MCMC samplers described in the methodology section. Each of the samplers are run such that 5000 samples are drawn with 1000 burn-in and tuning steps. We use leave-one-out(LOO) cross-validation error of [19] to evaluate the goodness of fit of each model.

Table 2 shows the LOO validation errors of the various models. It can be seen that the SIR model with two change points as the best model fit with the lowest mean LOO of 448.00. The

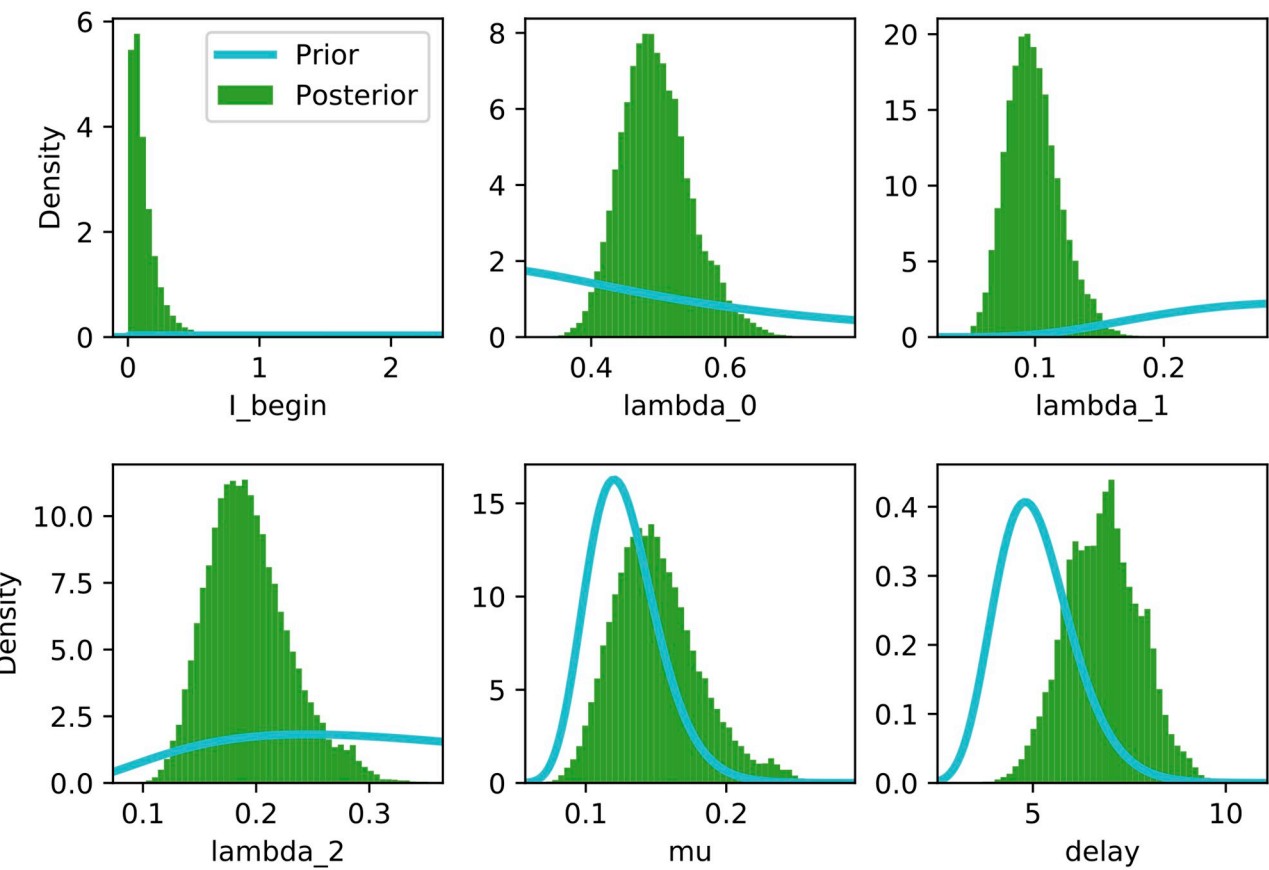

**Fig 4. Posterior parameter distributions for the SIR model with two change points.**

SEIR model with two change points showed a mean LOO of 459.94. We note that [6] similarly finds that the SIR model displayed superior goodness of fit to the SEIR on German data.

We now further present detailed results of the SIR and SEIR models with inference using NUTS, the trace plots from these models indicating stationarity in the sampling chains are provided in S2 and S5 Figs. The trace plots for the SIR and SEIR models using MH are provided in S3 and S6 Figs, while similar trace plots for slice sampling are provided in S4 and S7 Figs. The trace plots largely indicate that the NUTS sampler displays greater agreement between parallel chains thus lower rhat values.

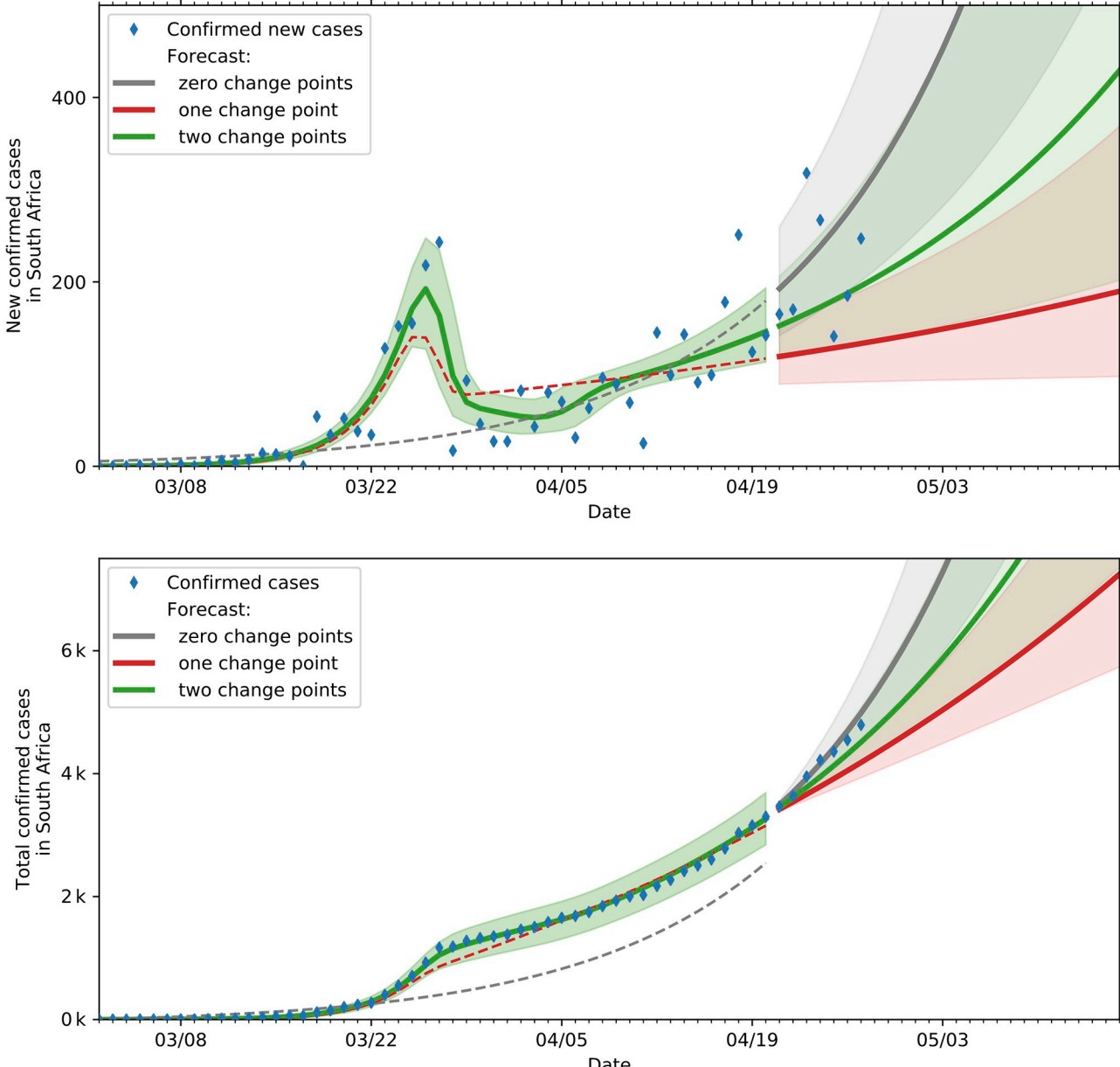

**Fig 5. Predictions and actual data(until 20 April 2020) based on SIR models with various change points.** The top plot indicates the actual and projected new cases while the bottom plot shows the actual and projected cumulative cases.

## Posterior parameter distributions

Fig 4 shows the posterior distributions of the SIR model parameters. The parameter estimates are $\lambda_0 \approx 0.495$ (CI[0.41, 0.564]), $\lambda_1 \approx 0.099$ (CI[0.065, 0.145]), $\lambda_2 \approx 0.197$ (CI[0.134, 0.264]), $\mu \approx 0.151$ (CI[0.09, 0.205]) and reporting delay $(D) \approx 6.848$ (CI[5.178, 8.165]). This corresponds to $R_0$ values of 3.278 (CI[2.715, 3.73]), 0.655 (CI[0.430, 0.960]) and 1.304 (CI[0.887, 1.7748]) at the respective change points. S1 Fig further shows the joint posterior distributions of $\lambda_t$ and $\mu$ at each of the change points.

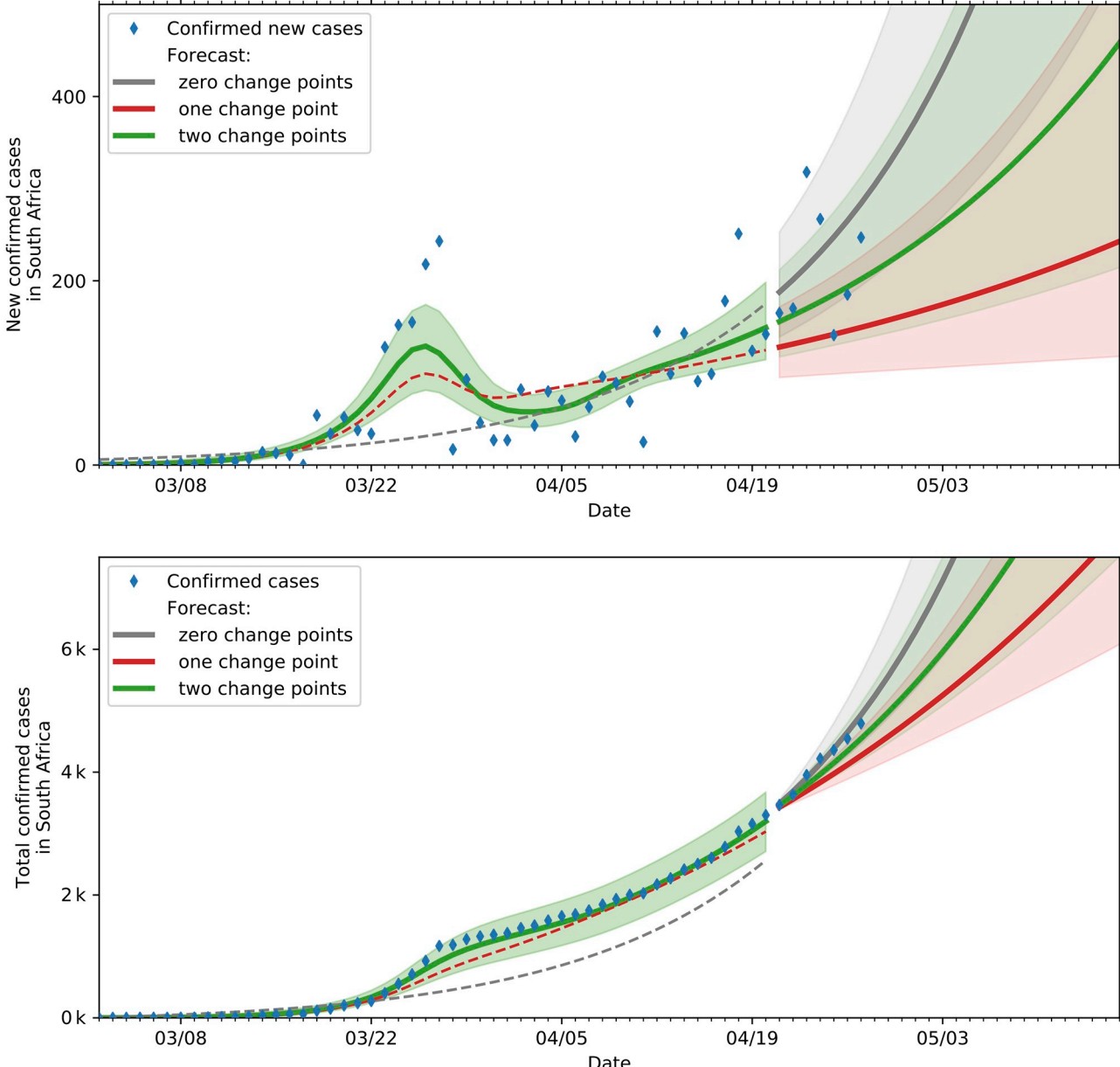

**Fig 6. Predictions and actual data(until 20 April 2020) based on SEIR models with various change points.** The top plot indicates the actual and projected new cases while the bottom plot shows the actual and projected cumulative cases.

Time-varying spread rates allow for inference of the impact of various state and societal interventions on the spreading rate. Fig 5 shows the fit and projections based on SIR models with zero, one and two change points. As can be seen from the plot the two change point model best captures the trajectory in the development of new cases relative to the zero and one change point models. The superior goodness of fit of the two change point model is also illustrated in Table 2. The fit and projections showing similar behaviour on the SEIR model with various change points are shown in Fig 6.

### Reporting delays, incubation and infectious period

The mean reporting delay time in days was found to be 6.848 (CI[5.178, 8.165]), literature suggests this delay includes both the incubation period and the test reporting lags. The posterior distribution incubation period from the SEIR model in Fig 7 yields a median incubation period of 4.537 days (CI[2.499, 6.787]). Thus suggesting a mean laboratory reporting delay of approximately 2.311 days. A mean recovery rate $\mu \approx 0.151$ implies mean infectious period of 6.620 days which is in line with related literature [2, 6, 10].

### Timing and impact of interventions

Fig 8 depicts the posterior distributions of the spreading rates and times corresponding to each change point. We observe that the first change point is on a mean date of 18 March 2020 (CI:[16/03/2020, 20/03/2020]). This date is consistent with the travel ban, school closures and social distancing recommendations. This change point resulted in a substantial decrease in the spreading rate (80%) primarily due to the reduction in imported infections.

The second change point is observed on 28 March 2020 (CI:[26/03/2020, 30/03/2020]). This time point coincides with the announcement of mass screening and testing by the

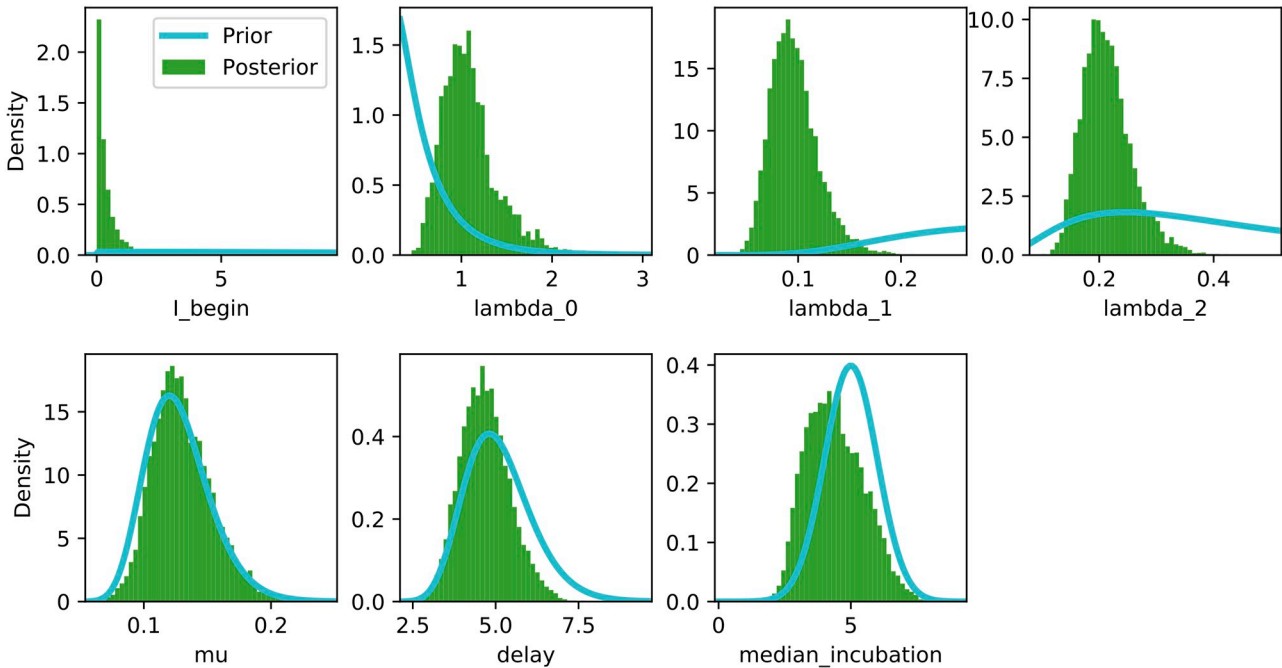

**Fig 7. Posterior parameter distributions under SEIR model with two change points.**

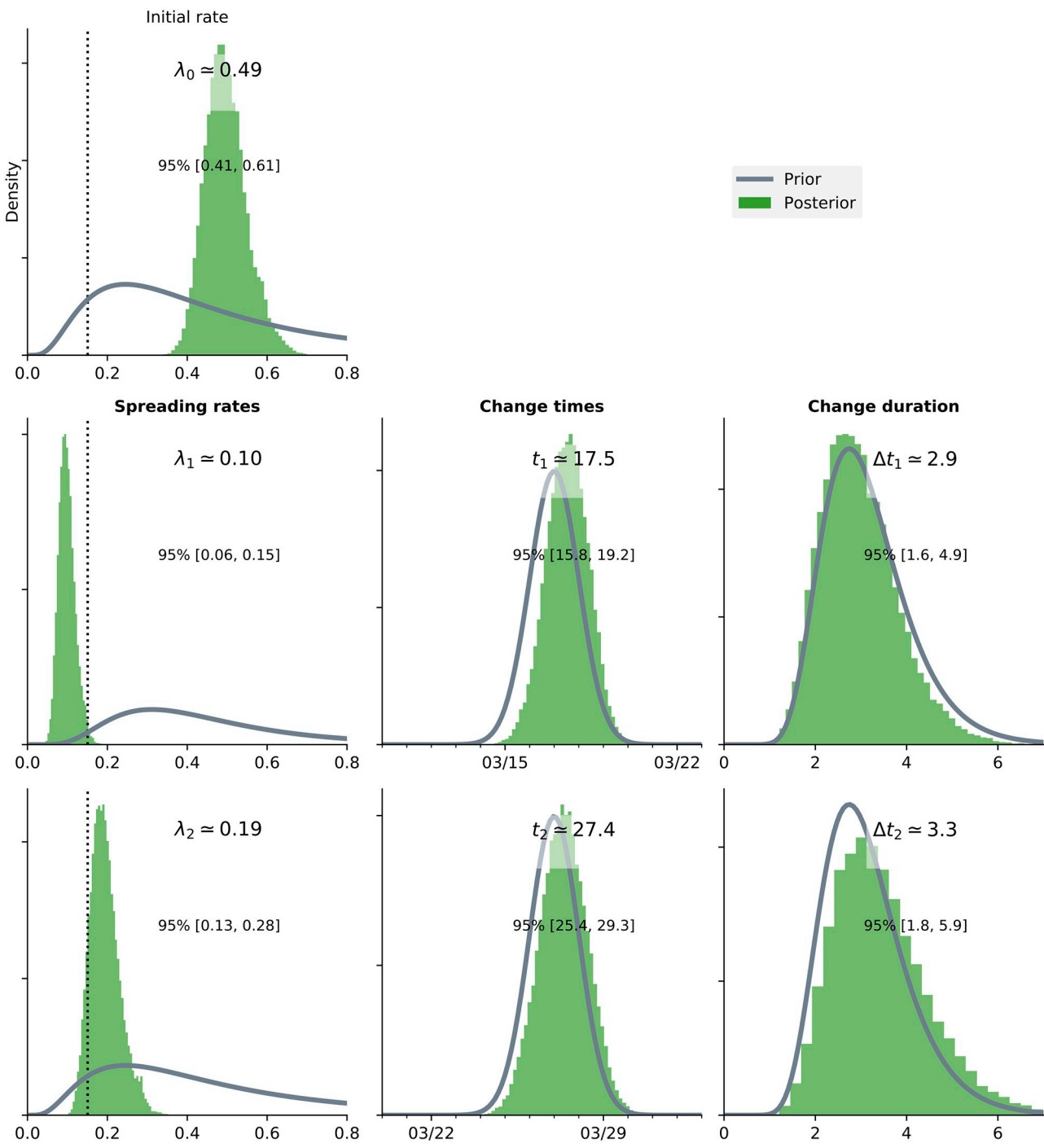

**Fig 8. Posterior distributions of the spreading rates($\lambda_t$) and the corresponding distributions of the time points.**

government on 30 March 2020. The resulting mean $R_0$ of 1.304 implies a 60% decrease from the initial value.

The inference of parameters is dependent on the underlying testing processes that generate the confirmed case data. The effect of the mass screening and testing campaign was to change the underlying confirmed case data generating process by widening the criteria of those

eligible for testing. While initial testing focused on individuals that either had exposure to known cases or travelled to known COVID-19 affected countries, mass screening and testing further introduced detection of community level transmissions which may contain undocumented contact and exposure to COVID-19 positive individuals.

## Discussion

We have performed Bayesian parameter inference of the SIR and SEIR models using MCMC and publicly available data as at 20 April 2020. The resulting parameter estimates fall in-line with the existing literature in-terms of mean baseline $R_0$ (before government action), mean incubation time and mean infectious period [2, 5, 6, 10].

We find that initial government action that mainly included a travel ban, school closures and stay-home orders resulted in a mean decline of 80% in the spreading rate. Further government action through mass screening and testing campaigns resulted in a second trajectory change point. This latter change point is mainly driven by the widening of the population eligible for testing, from travellers (and their known contacts) to include the generalised community who would have probably not afforded private lab testing which dominated the initial data. This resulted in an increase of $R_0$ to 1.304. The effect of mass screening and testing can also be seen in Fig 9 indicating a mean increase in daily tests preformed from 1639 to 4374.

The second change point illustrates the possible existence of multiple pandemics, as suggested by [20]. Thus testing after 28 March is more indicative of community-level transmissions that were possibly not as well documented in-terms of contact tracing and isolation

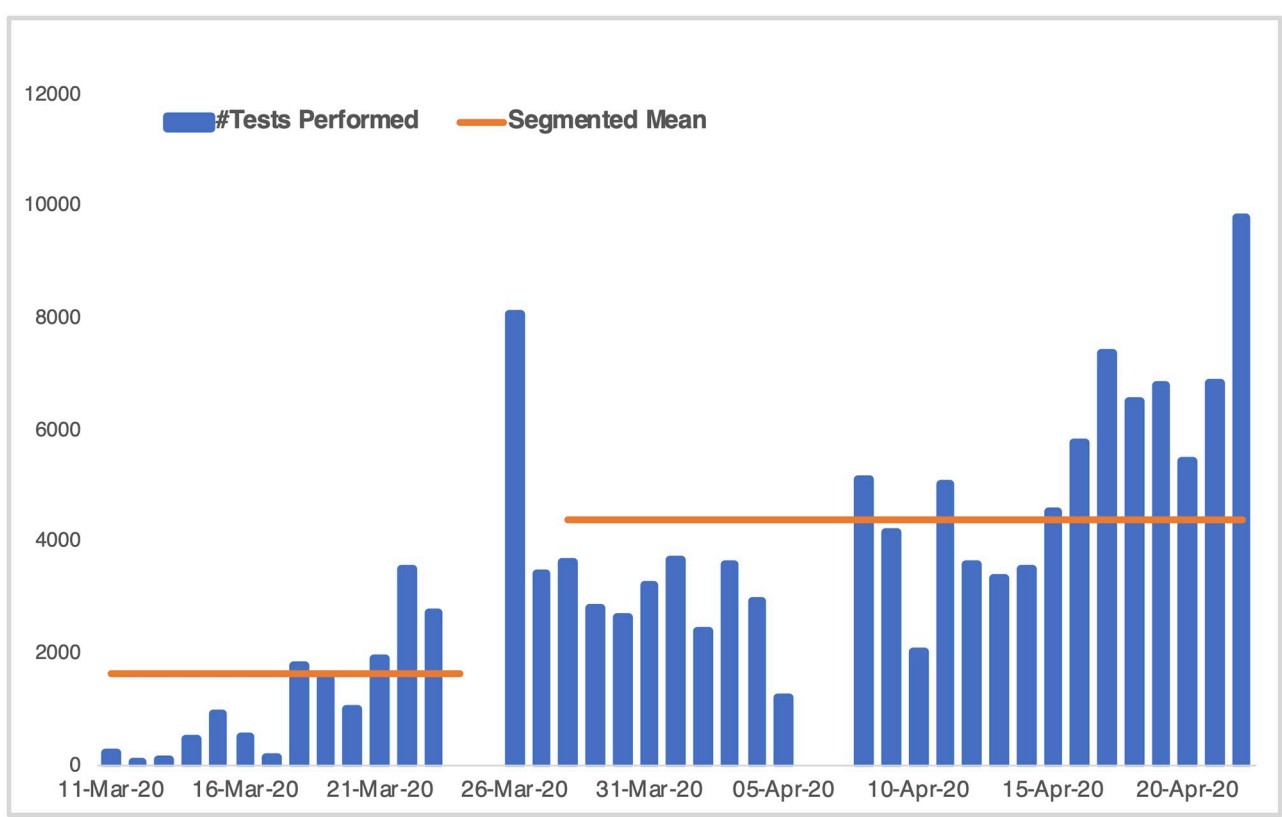

**Fig 9. Daily COVID-19 tests performed in South Africa.** The orange line indicates the segmented mean number of tests per day before and after the 28 March 2020 change point.

relative to the initial imported infection driven pandemic. This is also supported by the documented increase in public laboratory testing (relative to private) past this change point, suggesting health care access might also play a role in the detection of community-level infections [21].

## Conclusion

We have utilised a Bayesian inference framework to infer time-varying spreading rates of COVID-19 in South Africa. The time-varying spreading rates allow us to estimate the effects of government actions on the dynamics of the pandemic.

The results indicate a decrease in the mean spreading rate of 60%, which mainly coincides with the containment of imported infections, school closures and stay at home orders.

The results also indicate the emergence of community-level infections which are increasingly being highlighted by the mass screening and testing campaign. The development of the community level transmissions ($R_0 \approx 1.3041$ (CI[0.887, 1.7748])) of the pandemic at the time of publication appears to be slower than that of the initial traveller based pandemic ($R_0 \approx 3.278$ (CI[2.715, 3.73])).

A future improvement to this work could include extensions to regional and provincial studies as current data suggests varied spreading rates both regionally and provincially. As more government interventions come to play priors on more change points might also be necessary.

## Supporting information

**S1 Fig. Two dimensional heat maps of the joint posterior distributions of the spreading rate($\lambda$) and the recovery rate($\mu$) at various change points of the SIR model.** The high joint density areas (in yellow) indicate likely values of $R_0$. The baseline mean $R_0$ estimate in S1 Fig (a) is 3.278, the first change point estimate in Fig S1 Fig (b) is 0.655 while the second change point in S1 Fig (c) has resulted in a mean $R_0$ estimate of 1.304.
(TIFF)

**S2 Fig. Shows diagnostic trace plots for the SIR model inferred using NUTS.**
(TIFF)

**S3 Fig. Shows diagnostic trace plots for the SIR model inferred using MH.**
(TIFF)

**S4 Fig. Shows diagnostic trace plots for the SIR model inferred using slice sampling.**
(TIFF)

**S5 Fig. Shows diagnostic trace plots for the SEIR model inferred using NUTS.**
(TIFF)

**S6 Fig. Shows diagnostic trace plots for the SEIR model inferred using MH.**
(TIFF)

**S7 Fig. Shows diagnostic trace plots for the SEIR model inferred using slice sampling.**
(TIFF)

## Author Contributions

**Conceptualization:** Rendani Mbuvha, Tshilidzi Marwala.

**Data curation:** Rendani Mbuvha.

**Formal analysis:** Rendani Mbuvha.

**Investigation:** Rendani Mbuvha.

**Methodology:** Rendani Mbuvha.

**Supervision:** Tshilidzi Marwala.

**Visualization:** Rendani Mbuvha.

**Writing – original draft:** Rendani Mbuvha.

**Writing – review & editing:** Rendani Mbuvha, Tshilidzi Marwala.

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
