## [Decision Letter · Decision Letter 0]

11 Jun 2020

PONE-D-20-12397

Bayesian Inference of COVID-19 Spreading Rates in South Africa

PLOS ONE

Dear Dr. Mbuvha,

Thank you for submitting your manuscript to PLOS ONE. After careful consideration, we feel that it has merit but does not fully meet PLOS ONE’s publication criteria as it currently stands. Therefore, we invite you to submit a revised version of the manuscript that addresses the points raised during the review process.

Please respond to the reviewer comments on a point-by-point basis and revise the manuscript accordingly.

We look forward to receiving your revised manuscript.

Kind regards,

Jeffrey Shaman

Academic Editor

PLOS ONE

Journal Requirements:

2.We noticed you have some minor occurrence of overlapping text with the following previous publication(s), which needs to be addressed:

- arXiv:1906.06382v1

-https://www.medrxiv.org/content/10.1101/2020.04.07.20057133v2

In your revision ensure you cite all your sources (including your own works), and quote or rephrase any duplicated text outside the methods section. Further consideration is dependent on these concerns being addressed.

Please also ensure you have described the source of your data in the methods section, including a link or citation which can be used to access the data.

3. Please ensure that you refer to Figures 10,11 & 12 in your text as, if accepted, production will need this reference to link the reader to the figure.

4. Please upload a copy of Supporting Information Appendix S1 which you refer to in your text on page 11.

Reviewers' comments:

Reviewer's Responses to Questions

**Comments to the Author**

1. Is the manuscript technically sound, and do the data support the conclusions?

Reviewer #1: Yes

2. Has the statistical analysis been performed appropriately and rigorously? 

Reviewer #1: Yes

3. Have the authors made all data underlying the findings in their manuscript fully available?

Reviewer #1: Yes

4. Is the manuscript presented in an intelligible fashion and written in standard English?

Reviewer #1: Yes

5. Review Comments to the Author

Reviewer #1: In their paper the authors analyse with Bayesian methods the increase of COVID-19 case numbers in South Africa. They find two change points in the propagation that they link to respectively to travel bans/containment of infections and onset of massive testing. They conclude that the governmental interventions linked to the first change point were effective.

I recommend this paper for publication because of its timeliness and well described methods and would suggest a few minor revisions:

- In table 2 the authors list compare different sampling methods by their LOO statistic. As far as I understand it this is not useful and misleading. The LOO statistic was developed to compare different models (which are also correctly shown in this table). The lower LOO statistic of the MH sampling probably arises from a incomplete sampling of the posterior which could potentially be fixed letting the chains run longer and/or with a longer burn-in period. A useful statistic to check convergence is the Rhat statistic (https://docs.pymc.io/api/stats.html#pymc3.stats.rhat).

- The scale of the lambda_2 prior in the table 1 doesn't seem to match the prior distribution plotted in Fig. 4. The authors could also think about whether the lambda 1 and 2 priors are well motivated, and eventually make them wider/more uninformative. The posterior of the Lambda 1 prior is in the tail of the prior distribution (Fig. 4).

- In the algorithm of the HMC, in the third line of the leapfrog loop, the closing bracket after "p(t" should be removed.

6. PLOS authors have the option to publish the peer review history of their article (what does this mean?). If published, this will include your full peer review and any attached files.

Reviewer #1: Yes: Jonas Dehning

---

## [Author Response · Author response to Decision Letter 0]

7 Jul 2020

Dear Editor 

Kindly find our responses to reviews below.

Best,

Rendani Mbuvha

Responses to Academic Editor 

Response

We have made changes in line with the formatting guidelines, including:

 Changing the affiliation of one author from 'office of the vice-chancellor(his office)' to the institute that he is affiliated to at the university

Changing 'figure' references to 'fig'

Moving supporting information to an appropriately named appendix

Changing subsection headings accordingly

2.We noticed you have some minor occurrence of overlapping text with the following previous publication(s), which needs to be addressed:

- arXiv:1906.06382v1

-https://www.medrxiv.org/content/10.1101/2020.04.07.20057133v2

In your revision ensure you cite all your sources (including your own works), and quote or rephrase any duplicated text outside the methods section. Further consideration is dependent on these concerns being addressed.

Response:

We have made changes in terms of rewriting, rephrasing(with referencing) and inserting quotations in areas where there was previous overlapping text with our earlier preprints outside the methods section. Please see highlighted changes. 

Please also ensure you have described the source of your data in the methods section, including a link or citation which can be used to access the data.

Response:

We have added a description of our data source, including the necessary citation.

3. Please ensure that you refer to Figures 10,11 & 12 in your text as, if accepted, production will need this reference to link the reader to the figure.

Response:

These figures have now been moved to the supporting information files with the relevant references included in the main text.

4. Please upload a copy of Supporting Information Appendix S1 which you refer to in your text on page 11.

Response:

Now included as separate files - ( this mainly consists of the previous ‘figures’ 10-12 and other diagnostic plots)

Responses to Reviewer 1

I recommend this paper for publication because of its timeliness and well described methods and would suggest a few minor revisions:

- In table 2 the authors list compare different sampling methods by their LOO statistic. As far as I understand it this is not useful and misleading. The LOO statistic was developed to compare different models (which are also correctly shown in this table). The lower LOO statistic of the MH sampling probably arises from a incomplete sampling of the posterior which could potentially be fixed letting the chains run longer and/or with a longer burn-in period. A useful statistic to check convergence is the Rhat statistic (https://docs.pymc.io/api/stats.html#pymc3.stats.rhat).

Response:

We agree with the reviewer. We have revised table 2 to only refer to the comparison between models rather than include samplers. We have also increased the number of chain runs to 5000 and increased burn-in to 1000 runs with ten concurrent chains rather than four to increase the likelihood of convergence.

- The scale of the lambda_2 prior in the table 1 doesn't seem to match the prior distribution plotted in Fig. 4. The authors could also think about whether the lambda 1 and 2 priors are well motivated, and eventually make them wider/more uninformative. The posterior of the Lambda 1 prior is in the tail of the prior distribution (Fig. 4).

Response:

We have now changed both priors for lambda 1 and lambda 2 to a wider and relatively less informative LogNormal(log(0.4),0.7).

The change in priors and running the chains for longer marginally changes mean estimates of posterior parameters -- this does not seem to change the overall findings in a material way. 

- In the algorithm of the HMC, in the third line of the leapfrog loop, the closing bracket after "p(t" should be removed.

Response:

We have amended the formula typo accordingly.

---

## [Decision Letter · Decision Letter 1]

22 Jul 2020

Bayesian Inference of COVID-19 Spreading Rates in South Africa

PONE-D-20-12397R1

Dear Dr. Mbuvha,

We’re pleased to inform you that your manuscript has been judged scientifically suitable for publication and will be formally accepted for publication once it meets all outstanding technical requirements.

Kind regards,

Jeffrey Shaman

Academic Editor

PLOS ONE

Additional Editor Comments (optional):

Reviewers' comments:

Reviewer's Responses to Questions

**Comments to the Author**

1. If the authors have adequately addressed your comments raised in a previous round of review and you feel that this manuscript is now acceptable for publication, you may indicate that here to bypass the “Comments to the Author” section, enter your conflict of interest statement in the “Confidential to Editor” section, and submit your "Accept" recommendation.

Reviewer #1: All comments have been addressed

2. Is the manuscript technically sound, and do the data support the conclusions?

Reviewer #1: Yes

3. Has the statistical analysis been performed appropriately and rigorously? 

Reviewer #1: Yes

4. Have the authors made all data underlying the findings in their manuscript fully available?

Reviewer #1: Yes

5. Is the manuscript presented in an intelligible fashion and written in standard English?

Reviewer #1: Yes

6. Review Comments to the Author

Reviewer #1: The concerns have been fully addressed and in my opinion the manuscript is now acceptable for publication.

7. PLOS authors have the option to publish the peer review history of their article (what does this mean?). If published, this will include your full peer review and any attached files.

Reviewer #1: **Yes: **Jonas Dehning